# Effects of Dexmedetomidine on Immunomodulation and Pain Control in Videolaparoscopic Cholecystectomies: A Randomized, Two-Arm, Double-Blinded, Placebo-Controlled Trial

**DOI:** 10.3390/jpm13040622

**Published:** 2023-04-01

**Authors:** Gustavo Nascimento Silva, Virna Guedes Brandão, Marcelo Vaz Perez, Kai-Uwe Lewandrowski, Rossano Kepler Alvim Fiorelli

**Affiliations:** 1Department of Anesthesiology, Gaffrée e Guinle Universitary Hospital, Federal University of the State of Rio de Janeiro (UNIRIO), Rio de Janeiro 22290-240, RJ, Brazil; 2Department of Surgery and Anesthesia, Federal University of São Paulo (UNIFESP), São Paulo 04021-001, SP, Brazil; 3Center for Advanced Spine Care of Southern Arizona and Surgical Institute of Tucson, Tucson, AZ 85712, USA; 4Department of Orthopedics at Hospital Universitário Gaffrée e Guinle, Universidade Federal do Estado do Rio de Janeiro, Rio de Janeiro 20270-004, RJ, Brazil; 5Department of General and Specialized Surgery, Gaffrée e Guinle Universitary Hospital, Federal University of the State of Rio de Janeiro (UNIRIO), Rio de Janeiro 22290-240, RJ, Brazil

**Keywords:** laparoscopic cholecystectomy, immunomodulation, dexmedetomidine

## Abstract

Aim: Digital and robotic technology applications in laparoscopic surgery have revolutionized routine cholecystectomy. Insufflation of the peritoneal space is vital for its safety but comes at the cost of ischemia-reperfusion-induced intraabdominal organ compromise before the return of physiologic functions. Dexmedetomidine in general anesthesia promotes controlling the response to trauma by altering the neuroinflammatory reflex. This strategy may improve clinical outcomes in the postoperative period by reducing postoperative narcotic use and lowering the risk of subsequent addiction. In this study, the authors aimed to evaluate dexmedetomidine’s therapeutic and immunomodulatory potential on perioperative organ function. Methods: Fifty-two patients were randomized 1:1: group A—sevoflurane and dexmedetomidine (dexmedetomidine infusion [1 µg/kg loading, 0.2–0.5 µg/kg/h maintenance dose]), and group B—sevoflurane with saline 0.9% infusion as a placebo control. Three blood samples were collected: preoperatively (T0 h), 4–6 h after surgery (T4–6 h), and 24 h postoperatively (T24 h). The primary outcome was the level analysis of inflammatory and endocrine mediators. Secondary outcome measures were the time to return to normal preoperative hemodynamic parameters, spontaneous ventilation, and postoperative narcotic requirements to control surgical pain. Results: A reduction of Interleukin 6 was found at 4–6 h after surgery in group A with a mean of 54.76 (27.15–82.37; CI 95%) vs. 97.43 (53.63–141.22); *p* = 0.0425) in group B patients. Systolic and diastolic blood pressure and heart rate were lower in group A patients, who also had a statistically significantly lower opioid consumption in the first postoperative hour when compared to group B patients (*p* < 0.0001). We noticed a similar return to spontaneous ventilation pattern in both groups. Conclusions: Dexmedetomidine decreased interleukin-6 4–6 h after surgery, likely by providing a sympatholytic effect. It provides good perioperative analgesia without respiratory depression. Implementing dexmedetomidine during laparoscopic cholecystectomy has a good safety profile and may lower healthcare expenditure due to faster postoperative recovery.

## 1. Introduction

Laparoscopy represents one of the most innovative surgical techniques. It is associated with less surgical pain compared to open surgery, lower infection rates, shorter hospital permanence, and early return to work [1]. In developed countries, at least 10% of white adults have cholesterol gallstones. Women have twice the risk, and age increases the prevalence in both sexes. Symptomatic cholecystitis has reached epidemic proportions in the indigenous populations of North and South America and is associated with an increased risk of bile duct neoplasms. In contrast, the rate in sub-Saharan Africa and Asia is relatively low [2].

The insufflation of the peritoneum—pneumoperitoneum—allows safe laparoscopic surgery while improving direct surgical site visualization with less stress on the cardiovascular and respiratory systems. The increase in intraperitoneal pressure may compromise the regional blood flow, triggers a sympathetic adrenergic response with an increase in afterload, a reduction in venous return and preload that may result in a decreased cardiac output, and generate ischemia and reperfusion trauma after deflation [3]. The resultant systemic inflammatory response caused by the surgical trauma and the side effects of general anesthesia may contribute to the deterioration of postoperative respiratory function by generating diaphragmatic splinting and increased ventilatory pressures. Moreover, reduced ventilation/perfusion (V/Q) values from shunting may be induced by tracheal intubation resulting in decreased dead space due to bronchoconstriction, which occasionally occurs [4]. Surgical manipulation, tissue mobilization, excision, and dissection can lead to elevated levels of inflammatory mediators, and cytokine release may drive organ dysfunction via immunological, metabolic, and hormonal processes [5,6]. Therefore, multimodal anesthetic interventions in surgery with strategic applications of modern drugs and regional blocks are needed to decrease the frequency of perioperative adverse events, improve recovery with earlier return to regular activities of dialing living, and reduce healthcare expenditures [7].

Dexmedetomidine is a selective and potent α2-adrenergic receptor agonist. It acts on the activation of supraspinal pre- and postsynaptic receptors in the spinal cord’s locus coeruleus and dorsal horn cells [8]. It is believed that its association with general anesthesia may promote control of the trauma response by altering the neuroinflammatory reflex through the antinociception and immunomodulation pathways. By attenuating the excessive release of noradrenaline during ischemia-reperfusion, with decreased potential to form reactive species of oxygen, some authors have reported a superior anti-inflammatory effect than other drugs. Antiapoptotic activity, better modulation of macrophage function, reduction of pro-inflammatory mediators such as Interleukin 6 (IL-6) and tumor necrosis factor α (TNF-α), and inhibition of the inflammatory reaction to sepsis have also been reported [9,10,11]. Therefore, our group used a safe target-controlled continuous infusion protocol to evaluate the clinical benefits of dexmedetomidine administration by analyzing perioperative levels of inflammatory and endocrine mediators as a measure of surgical stress and organ function.

## 2. Materials and Methods

### 2.1. Study Design, Informed Consent and Trial Registration

This double-blinded prospective randomized study included 52 patients of both genders scheduled for elective laparoscopic cholecystectomy. Enrolled patients signed informed consent. The recruitment was conducted after approval by the Ethics and Research Committee of the Gaffrée e Guinle University Hospital, Federal University of the State of Rio de Janeiro (UNIRIO), Rio de Janeiro, Brazil, in December 2021 (CAAE No. 50311621.0.0000) and registered on the Brazilian Clinical Trial Registration Platform (REBEC, https://ensaiosclinicos.gov.br/, RBR-2rgxbdv, last accessed on 19 January 2023), on 28 March 2022. The Surgery and Anesthesiology Departments are credentialed by the Federal Ministry of Education and the Brazilian Society of Anesthesiology. The study was conducted at Gaffrée e Guinle University Hospital, Federal University of the State of Rio de Janeiro (UNIRIO), where all surgeries were performed between April and October 2022.

### 2.2. Inclusion/Exclusion Criteria

Patients aged between 18 and 70 years and a physical status classification using the American Society of Anesthesiologists (ASA) between ASA 1 and 2 needing surgical treatment for symptomatic cholecystitis were enrolled. Patients were excluded from the study if their surgery was converted to open surgery for any reason considering the presumed increase in surgical trauma and the absence of pneumoperitoneum ischemia-reperfusion syndrome. Additional exclusion criteria were surgery duration of greater than 3 h, illicit drugs use, prescription medication use known to induce the cytochrome complex P450, patients with renal and hepatic failure, chronic users of corticoids and non-steroidal anti-inflammatories (NSAIDs), body mass index (BMI) greater than or equal to 40, heart failure, valvular or ischemic heart disease, and use of tricyclic antidepressants.

### 2.3. Double-Blinded Randomization Protocol

Consented study participants were randomized using a list of random numbers in varying block sizes on a 1:1 ratio with no restrictions to any of the two study groups: Sevoflurane and Dexmedetomidine group—A vs. Sevoflurane and Saline 0.9%—B (Figure 1). Randomization was performed by the nursing team together with the pharmacy sector through a schedule provided by the statistics staff. Patient data were replaced by computer-generated numbers, and these codes were inserted in sealed envelopes drawn before the infusion of the solution by the surgeon. The pharmacist prepared the medicine or placebo, labeled it with the study subject code, and physically delivered it by the pharmacy staff to the anesthesiologist (G.N.S) performing the infusion. The patients and clinical investigators of the surgical and anesthesia team were blinded to the patient’s group assignment. The nursing and pharmacy staff preparing, handling, and delivering the drug preparations to the operating room were not involved in any part of the research. The study adhered to the applicable CONSORT (Consolidated Standards of Reporting Trials) guidelines.

### 2.4. Anesthetic Procedures

All patients had peripheral venous access with 20 or 18-gauge peripheral intravenous catheters. There were no differences in administering the anesthesia protocol between the two groups. Induction of general anesthesia was achieved through intravenous administration with fentanyl (5 mcg/kg), lidocaine (1.5 mg/kg), propofol (2.5 mg/kg), and cisatracurium (0.15 mg/kg) until TOF equals zero. Positive pressure ventilation was started after endotracheal intubation with a 6–8 mL/kg tidal volume. The respiratory rate was titrated to maintain an end-tidal CO_2_ between 35 and 45 mmHg. Intraoperatively, general anesthesia was maintained with sevoflurane 1 to 1.5 MAC with 30% oxygen and 70% air. The bispectral index was held between 40 and 60 during the operation. All patients received dipyrone at 30–50 mg/kg and ketoprofen at 100 mg intravenous at the end of surgery. The surgical portals were infiltrated with 15 mL of ropivacaine 0.3% for improved analgesia. Additional doses of opioids were not given intraoperatively but ondansetron 8 mg for nausea and vomiting prophylaxis. We proceeded to extubation after recovery of neuromuscular function by TOF > 90%. In the recovery room, patients received intravenous opioids (morphine 0.05–0.1 mg/kg, intravenous) if in moderate to severe pain. The dexmedetomidine (1 mcg/mL) was administered by target-controlled continuous infusion, after venoclysis, at one mcg/kg for 20 min, followed by 0.2–0.5 mcg/kg/h until surgical closure. A placebo solution with 0.9% saline was infused in the B Group.

Variations in the hemodynamic parameters between the two groups were compared through analysis of recordings of heart rate (HR), systolic blood pressure (SBP), and diastolic blood pressure (DBP) at the following intervals: preoperatively (time 0—T0), during anesthetic induction (and subsequent 25 min), surgery, and wake up. Three blood samples were collected: T0 h—before surgery in the preoperative holding area, 4 to 6 (T4–6 h), and 24 h (T24 h) after surgery just before discharge from the hospital.

### 2.5. Laboratory Assay

Interleukin-6 (IL-6), cortisol, C-reactive protein, and glucose were measured in venous blood samples, centrifuged at 2000 rpm for 15 min, and stored in the biorepository at 80 °C in cryotubes in the Immunology and AIDS Research Laboratory at the University Hospital. The tests were done at the National Quality Control Program (NQCP) laboratory in Rio de Janeiro six months after the study initiation. The serum concentration of IL-6 was determined by electrochemiluminescence immunoassay using the Roche Cobas e411^®^ immunoassay analyzer. The serum concentration of CRP was determined by a Roche^®^ immunoturbidimetric assay using the Bioclin 3000 automated analyzer. Blood glucose values were determined by colorimetric enzymatic assay—GOD-PAP (Trinder) Roche^®^, using the automatic Bioclin 3000 analyzer. The serum cortisol concentration was determined by chemiluminescence immunoassay (CLIA) Roche^®^, using the Abbott Architect i1000 immunoassay analyzer.

### 2.6. Outcome Measures

The primary outcome measure was evaluating the inflammatory response to the surgical trauma during the laparoscopic cholecystectomy via analysis of the changes in the Interleukin 6 (IL-6), Cortisol, C-reactive protein (CRP), and glucose level as a result of dexmedetomidine administration. Secondary outcomes were indirect measures of surgical stress and trauma by assessing lung function before extubation was evaluated by the return to spontaneous ventilation test with analysis of current volume values, respiratory rate (RR), peripheral oxygen saturation (SpO_2_), and expired carbon dioxide fraction [12]. SpO_2_ and expired CO_2_ fractions were used as oxygen and carbon dioxide tension indicators. Data were recorded when the exhaled fraction of sevoflurane reached values less than or equal to 0.3 MAC and patients started spontaneous breathing before extubation. Hemodynamic changes between the two groups were compared through changes in graphics behavior in heart rate (HR), systolic blood pressure (SBP), and diastolic blood pressure (DBP). The quality of analgesia was evaluated by two factors:the amount of opioid (morphine) used at one hour, six hours, and 24 h after surgery and the pain reported by the patient using the visual analog pain scale (VAS).

### 2.7. Statistical Power & Analysis

This study was designed to detect a difference in perioperative plasma level of endocrine and inflammatory mediators between the groups, aiming to reach an adequate Cohen effect size for the variables. The sample size was estimated to compare means and proportions, with a statistical power of 80% for all variables and a significance level of 5% (α = 0.05). Considering the patient’s eligibility criteria and the desired power, we determined a minimum sample size of 52 patients (26 patients per group), similar to a study with an approximate design [13].

The T-test was chosen for the variables that follow a normal distribution for the control and intervention groups by the Shapiro–Wilk test. The non-parametric Mann–Whitney test was used to compare medians and quantitative variables. The non-parametric Chi-square test was chosen considering the categorical variables, employing a 95% confidence interval in the analysis. Statistical analyses were performed using R software version 4.0.3 (R Project for Statistical Computing, Vienna, Austria).

## 3. Results

### Study Participants

Fifty-nine patients were initially selected for the study, but seven were excluded. Two patients were excluded because their surgery took longer than three hours, two due to assignment to open surgery for other non-study related reasons, and the remaining three due to obesity with a BMI greater than 40. The study was concluded once 52 eligible patients had been assigned between the two groups—sevoflurane and dexmedetomidine group A (*n* = 26)—vs. sevoflurane and saline 0.9% group B (*n* = 26). Our study’s CONSORT flow diagram is shown in Figure 1.

The demographic and clinical characteristics of the patients included in the study are shown in Table 1. The groups were matched for age, sex, height, weight, BMI, comorbidities, and surgical indication. The results of the non-demographic variables are contained in Table 2.

Surgical time was similar between groups, lasting less than three hours. The occurrence of pain and opioid use was higher in the first hour after surgery in the control group. None of the patients developed bradycardia or significant hemodynamic instability requiring suspension of medication infusion in the intraoperative period. A buffering for IL-6 at four to six hours after surgery was found in the intervention group, demonstrating the biological effect of alpha-2 blockade on immune response with statistical significance (Figure 2).

Postoperatively, CRP remained elevated in both groups, and dexmedetomidine alone was insufficient to change the endocrine response to surgical trauma. There was no difference between the groups regarding postoperative cortisol and blood glucose levels (Table 2). Hemodynamic variables, including SBP, DBP, and HR, revealed that patients in the interventional group A had lower blood pressures, thus, corroborating the sympatholytic effect of dexmedetomidine from induction to wake up (Figure 3). The administered doses did not cause significant hypotension and bradycardia to the point that there was a need to stop its infusion. The groups had no statistical difference in the use of amines (ephedrine).

The *p*-values calculated using the Chi-square test for operative outcome variables were significant for opioid necessity and the category of postoperative pain one hour after the procedure. For the one-hour time point, the groups showed a statistical difference in opioid consumption (*p* < 0.0001) and pain score (*p* < 0.0001) with lower opioid use and lower pain scores in the A group (Figure 4). There was no difference between the groups for the four and 24-h time points.

The RR and SpO_2_ parameter analysis suggested a statistically significant faster return to spontaneous ventilation after extubation in patients treated with dexmedetomidine: group B RR = 13.96 (13.18–14.74) vs. group A = 15.62 (14.47–16.62; *p* = 0.0361) and SpO_2_, B = 0.97 (0.97–0.98) vs. A = 0.98 (0.98–0.99; *p* = 0.0002). Our results showed that treatment with dexmedetomidine did not provide deterioration of respiratory function, maintaining adequate parameters similar to the control group.

## 4. Discussion

The adrenergic tone on the immune system determines endocrine-metabolic changes and demonstrates the intercommunication between the neural, glandular effector and immune systems [14]. Although immunosuppression in the perioperative period may increase the risk of infections, the anti-inflammatory effects of some medications may promote benefits in controlling systemic inflammatory response syndrome (SIRS) and imply a favorable immediate postoperative outcome and early hospital discharge [15]. The anti-inflammatory mechanisms of dexmedetomidine are being widely studied. Three hypotheses could explain this effect: (1) regulation of cytokine production by immune system cells, (2) antinociception, and (3) alteration of the cholinergic anti-inflammatory pathway by central sympatholytic effect [8,10,16]. Innate immunity cells are capable of expressing alpha-2 adrenoreceptors on their cell membrane. Blocking adrenergic tone on effector tissues can contribute to the modulation of cytokine production by lymphocytes, macrophages, and monocytes during the stress response and reductions in serum levels of pro-inflammatory cytokines such as IL-6, IL-8, and TNF-α throughout up to 24 h after surgery. The control of immune and inflammatory reactions, with fine tuning between pro- and anti-inflammatory cytokines, is essential to minimize significant pathological damage in various settings such as trauma, sepsis, and cancer [9,17,18].

In our study, IL-6 responses were associated with the operative injury’s magnitude and the operative procedure’s invasiveness. This marker may be helpful in the objective assessment in determining the impact of their levels to improve patient outcomes and to assess the possible immune function modulation [19]. This interleukin is a marker of the inflammatory response to surgical trauma that induces synthesis of acute phase reactants by the liver, stimulates neutrophil production in the bone marrow, and promotes differentiation of T helper cell producers of IL-17. It is produced by macrophages, dendritic cells, endothelial cells, fibroblasts, and other cells in response to pathogen-associated molecular patterns (PAMPS), IL-1, and TNF. The innate lymphoid cells of group 2 are activated in response to the epithelial cell-derived cytokines IL-33 and IL-25. These cells release mediators associated with a Th2 response, such as IL-6. Its serum increase directly reflects the magnitude of stress, as in sepsis, resulting in mitochondrial dysfunction, glycocalyx disruption, and endothelial dysfunction, implying increased morbidity and mortality [13,20,21]. Thus, decreasing IL-6 release may signify control of the surgical stress response. We found a buffering effect depicted in the graph for IL-6, 4–6 h after surgery in the intervention group, demonstrating the biological impact of alpha-2 blockade on immune response with statistical significance: in group A, 54.76 (27.15–82.37) vs. 97.43 (53.63–141.22); *p* = 0.0425 in group B.

Studies have shown that dexmedetomidine did not provide an adequate protective effect on stress hormones (epinephrine, cortisol) [22,23]. This may be due to its primary mechanism of action being via hyperpolarization of noradrenergic neurons of the locus ceruleus [24]. However, combined with other adjuvants, such as propofol in a continuous infusion, it can effectively alleviate the stress response of patients undergoing laparoscopic cholecystectomy and potentiate perioperative hemodynamic stabilization [25]. Our results also suggested that intervention with dexmedetomidine as a single agent was insufficient to alter the endocrine response to surgical trauma. There was no difference between groups when cortisol and glycemia serum dosages were compared in the postoperative period. Controlling the endocrine response is essential for managing postoperative outcomes after trauma. Metabolic and hydro electrolytic changes resulting from the adrenergic response on the effector endocrine tissue can precipitate harmful events in a susceptible organism. Therefore, multimodal anesthesia, with the strategic use of drugs with different mechanisms of action and regional blocks, is crucial when this goal is pursued [26,27,28].

Activation of the anti-inflammatory cholinergic pathway is a survival mechanism to attenuate sympathetic effects during surgical trauma. Via the vagus nerve, the reflex’s afferent fibers detect inflammatory mediators and transmit these signals to the dorsal motor nucleus, generating an efferent signal and release of ACh on α-7 nicotinic receptors on the surface of macrophages and cells of innate immunity. This vagal pathway inhibits the release of pro-inflammatory mediators, including IL-1β, IL-18, and TNF-α. After surgical trauma, the body must balance physiological processes, such as wound healing, and the cholinergic anti-inflammatory response by decreasing exacerbated systemic inflammation [29,30]. In this context, a sympatholytic effect may reduce pro-inflammatory mediators and cardiovascular stability in patients admitted to intensive care units. In our study, SBP, DBP, and HR values were significantly lower in group A patients.

Analgesic properties of dexmedetomidine can be attributed to its spinal, supraspinal, and peripheral actions. Acting on alpha-2 adrenergic receptors, mainly on alpha2A subtypes, promotes neuronal hyperpolarization and reduction of calcium channel activation. It contributes to the decrease of hyperalgesia and allodynia, modulating the maintenance mechanism of chronic pain [31,32].

The abusive use of opioids has become a worldwide public health crisis. Between 2001 and 2006, opioid-related deaths in the United States increased by 345%. Therefore, its controlled use in the postoperative period is paramount [33]. For the one-hour time point, our study groups showed a statistical difference in opioid consumption (*p* < 0.0001) and pain score (*p* < 0.0001) with less opioid use and lower pain scores in the dexmedetomidine intervention group (Table 2/Figure 4). Several studies have demonstrated the efficacy of dexmedetomidine in controlling postoperative pain. It is associated with improved quality of postoperative recovery and reduced opioid consumption in the immediate postoperative period. These factors make dexmedetomidine an attractive agent for enhanced recovery in surgery (ERAS) protocols and for patients with acute and chronic pain [34,35,36,37]. Pain management further considers it a strategy to decrease opioid use/abuse and as an adjuvant with other drugs in regional peripheral and neuroaxis blocks to increase the duration and quality of analgesia [38,39].

Research comparing dexmedetomidine with remifentanil infusion showed that both have equal efficacy in attenuating cough in patients undergoing cerebral aneurysm clamping. However, dexmedetomidine provided better preservation of respiratory function. Unlike the injection of opioids or benzodiazepines, dexmedetomidine can be infused safely, aiming for adequate tracheal extubation. It can protect against adverse respiratory events in specific situations, such as awake craniotomy and intubation, without promoting residual sedation and respiratory depression [40,41,42]. Similarly, in the sedation regimen of critical pediatric patients, the safety and efficacy profile in reducing the incidence of a withdrawal syndrome after weaning from analgesic and sedative drugs in pediatric ICU dexmedetomidine appeared to confer promising results [43]. During extubation, the study showed that dexmedetomidine provided similar quality in the return of respiratory function upon awakening in both groups. Selective and potent agonism to the α2-adrenergic receptor is responsible for these anxiolytic and sedative properties. Activation of supraspinal pre- and postsynaptic α2-adrenergic receptors in the locus coeruleus may influence endogenous sleep-promoting pathways, contributing to potent and effective sedatives/analgesics effects [8,44].

## 5. Limitations

The study had several limitations. First, it failed to evaluate dose-dependent changes in the protection of specific organs employing markers of protection, such as the antiapoptotic regulatory pathways of postoperative cognitive dysfunction. Second, this study did not analyze the effects of different doses of dexmedetomidine on the anesthetics and analgesics consumption during surgery. Third, the sample size may have prevented achieving statistical differences and power of 80% for other variables analyzed. However, the reduced volume of elective surgeries during the COVID-19 pandemic and the low availability of supplies indirectly affected the sampling. Fourth, RCP was not measured 36–48 h after surgery and may have limited reaching the peak values with possible drug interference.

## 6. Conclusions

The inflammatory response to trauma is influenced by numerous factors, with the neuroinflammatory factor being paramount. However, it is often neglected. In this trial, we found that the association of dexmedetomidine with general anesthesia for laparoscopic cholecystectomy damped the inflammatory response by decreasing the release of IL-6, the primary pro-inflammatory mediator, in the immediate postoperative period. This investigation demonstrated the benefits of using alpha-2 agonists in the perioperative period due to their analgesic and sympatholytic effects, with respiratory and cardiovascular safety. Its use is linked to less opioid consumption in times of overuse in the immediate postoperative period. There was better control of critical clinical parameters in managing aseptic trauma, which can be considered and indicated in this surgical intervention. Finally, the importance of a better understanding of its molecular mechanisms of organ protection, targeted by promising studies, is emphasized for the future.

## Figures and Tables

**Figure 1 jpm-13-00622-f001:**
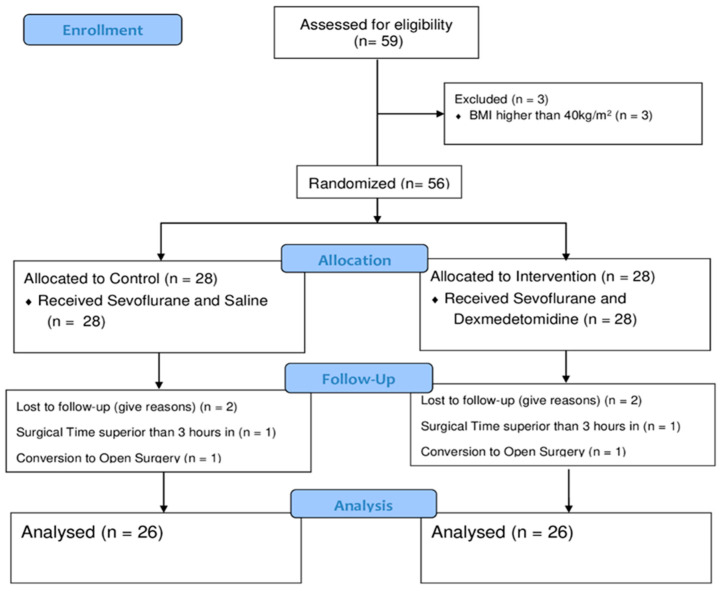
Consort flow diagram.

**Figure 2 jpm-13-00622-f002:**
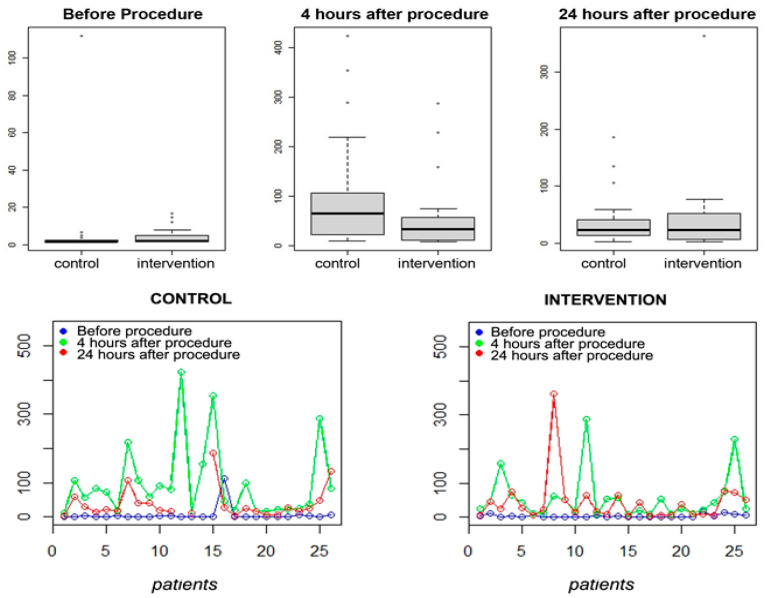
Boxplots and Graphs from the variable IL-6. Note that the median, represented by the central line of each boxplot, at four to six hours after surgery, showed more distant values between the control and intervention groups than for the other periods recorded. At the 5% significance level, there was a statistical difference between the groups. Graphs: values observed by each patient for IL-6 measurement time between groups.

**Figure 3 jpm-13-00622-f003:**
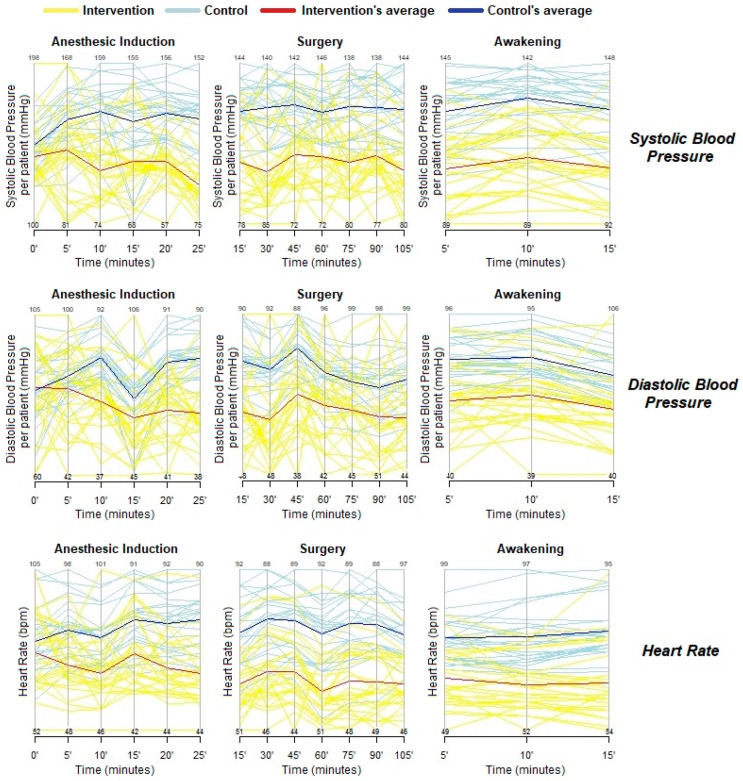
Hemodynamic behavior.

**Figure 4 jpm-13-00622-f004:**
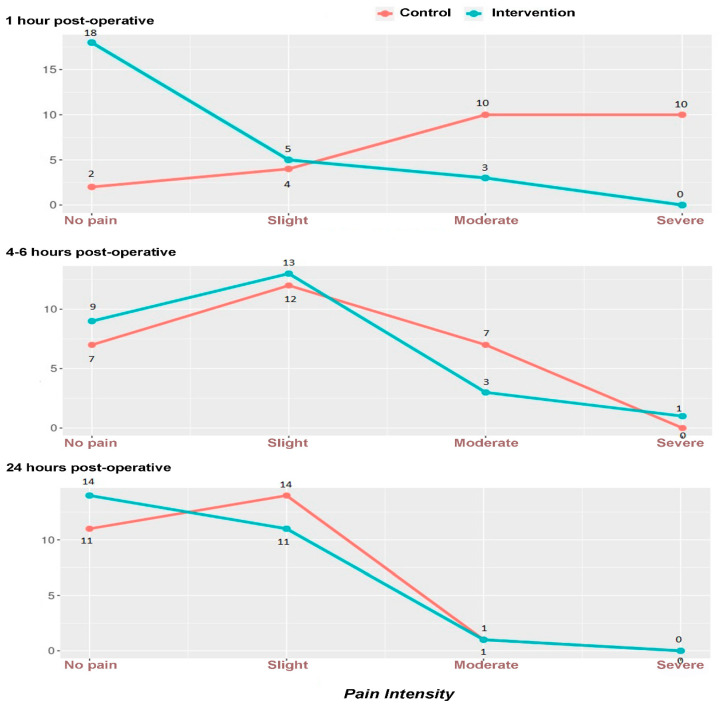
Graphs of the incidence of postoperative pain using the categories no pain, slight, moderate, and severe pain.

**Table 1 jpm-13-00622-t001:** Demographics and Clinical Data.

	Control (n = 26)	Intervention (n = 26)	*p*-Value
Sex (Male/Female)	8/18	6/20	0.7546 *
Age (years)	53.23 (43.3–63)	48 (42.3–60)	0.3598 **
Height (cm)	162.5 (159–170.2)	164.5 (160.3–169.8)	0.6535 **
Weight (kg)	71.5 (65.8–79.8)	76 (68–90)	0.3993 **
BMI	26.6 (25.5–29.4)	29.9 (24.1–31.6)	0.6049 **
ASA Status (I/II)	8/18	7/19	1.0 *
Surgical Indication (Biliary Polyposis/Biliary Lithiasis)	2/24	1/25	1.0 *
Dose of Dexmedetomidine (mcg)	0	120.5 (93–159)	**-**

ASA = American Society of Anesthesiologists/BMI = Body Mass Index. Data are reported as mean (95% CI of mean) or absolute frequency. * Chi-square test/** Mann–Whitney test.

**Table 2 jpm-13-00622-t002:** Results from hypothesis tests for the non-demographic variables of the study.

**Inflammatory Biomarkers**			
	**Control (n = 26)**	**Intervention (n = 26)**	***p*-Value**
	**Mean (95% CI)**	**Mean (95% CI)**	
IL-6			
0 h	6.37 (2.33–15.07)	4.09 (2.39–5.79)	0.2090 **
4 h	97.43 (53.63–141.22)	54.76 (27.15–82.37)	0.0425 **
24 h	37.86 (19.13–56.58)	43.03 (14.81–71.25)	0.9149 **
Cortisol			
0 h	11.16 (8.85–13.53)	11.77 (9.16–14.38)	0.8489 **
4 h	22.92 (18.15–27.69)	22.12 (16.98–27.26)	0.6565 **
24 h	12.52 (8.53–16.51)	11.58 (9.03–14.14)	0.8852 **
C Reactive Protein			
0 h	2.91 (1.74–4.08)	7.28 (4.45–10.11)	0.0067 **
4 h	4.61 (2.89–6.32)	9.41 (6.58–12.24)	0.0075 **
24 h	39.48 (29.86–49.09)	44.21 (33.21–55.21)	0.6481 **
Glycemia			
0 h	91.19 (83.42–98.96)	83.46 (74.52–92.40)	0.0255 **
4 h	111.19 (101.54–120.84)	111.07 (103.70–118.45)	1.0 **
24 h	110.29 (101.06–119.52)	113.38 (100.08–126.68)	0.9535 **
**Perioperative Outcome**			
	**Control (n = 26)**	**Intervention (n = 26)**	***p*-Value**
	**Mean (95% CI)**	**Mean (95% CI)**	
Opioide (yes/no)	18/8	2/24	<0.0001 *
Postoperative Pain (No pain/Slight/Moderate/Severe)			
1 h	2/4/10/10	18/5/3/0	<0.0001 *
4 h	7/12/7/0	9/13/3/1	0.4089 *
24 h	11/14/1/0	14/11/1/0	0.6977 *
**Spontaneous Ventilation Return**			
	**Control (n = 26)**	**Intervention (n = 26)**	***p*-Value**
	**Mean (95% CI)**	**Mean (95% CI)**	
RR	13.96 (13.18–14.74)	15.62 (14.47–16.62)	0.0361 **
TV(ML)	400.04 (381.89–418.19)	372.85 (348.92–396.77)	0.1325 **
EFCO_2_	40.42 (38.67–42.17)	41.04 (39.49–42.59)	0.5361 **
SpO_2_	0.97 (0.97–0.98)	0.98 (0.98–0.99)	0.0002 **

IL-6 = Interleukin-6/ SpO_2_ = peripheral oxygen saturation /RR = Respiratory Rate/TV = Tidal Volume/EFCO_2_ = expired fraction of carbon dioxide. Data are reported as mean (95% CI of mean) or absolute frequency. * Chi-square test/** Mann–Whitney test. Reported measurement units: IL-6—pg/mL; RCP—mg/L; Glycemia—mg/dL; Cortisol—mcg/dL.

## Data Availability

The data are contained within the article.

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
