# Peer review of "Effects of Dexmedetomidine on Immunomodulation and Pain Control in Videolaparoscopic Cholecystectomies: A Randomized, Two-Arm, Double-Blinded, Placebo-Controlled Trial"

_jpm, 2023, doi:10.3390/jpm13040622_

Round 1

Reviewer 1 Report

The pharmacodynamic effects of dexmedetomidine are well known and this study doesn't add anything to our knowledge. It is a very small cohort and the power analysis makes no sense to me. Anesthesia claims to be standardised but there is no information on other drugs used e.g. NSAID, dexamethasone, anti-emetics - all may influence inflammatory response. Pain scores were a bit less in the first hour, but the difference is not clinically significant given that the surgery is not that painful anyway and not one expected to produce a major inflammatory response. The use of surrogate markers is not helpful as these are meaningless to patients having this type of surgery - other parameters of quality of recovery would be more important  

Author Response

Author's Notes:

     We are glad to know that, based on the advice received our manuscript entitled “Effect of Dexmedetomidine on Immunomodulation and Pain Control in Videolaparoscopic cholecystectomies: A Randomized, Two-Arm, Double-Blinded, Placebo-Controlled Trial" has merit for publication. We thank you for your time spent evaluating our manuscript and for the valuable input given to improve it.

        We send a updated version for your appreciation.

Reviewer 2 Report

Thank you to the authorship team for this important manuscript exploring the effect of intraoperative dexmedetomidine on immunomodulation and pain in laparoscopic cholecystectomies. As stated by the authors, this is an important research question for which there is paucity of agent-specific data as well as correlation to clinical outcomes. 

Abstract

Line 22 and 28 - 'quality upon awakening' I would challenge that this term would be broader than the outcomes actually measured and could be misleading. Namely, only pain and respiratory parameters were measured. 

Line 30 - 'dexmedetomidine provided anti-inflammatory activity' This statement seem broader and more definitive than the results. Namely, a reduction in IL-6 seen at 4 hours but not at later time points or reflected in other neurohormonal markers measured.

Introduction

Line 48/49 'an increase in dead space leading to bronchoconstriction. Do the authors mean the reverse?

Line 50-56. The authors are attempting to make a link between inflammatory processes and the need to ERAS protocols. I am unaware of a definitive correlation between these two concepts

Line 63 - Could the authors clarify 'medium-sized surgical models'

Line 69- References 12-14 don't appear to relate to the content written

Methods

Could the authors comment to dexmedetomidine dosing? Was this protocolised? If so, how? If haemodynamics are measure at induction it would be important as to the timing to administration

Could the authors comment as to if fraction of inspired oxygen was protocolised and also opioid administration as this seems relevant to oxygen saturations and respiratory rate

Results

Could the authors comment as to how patients whose surgery was greater than 3 hours, or who were converted to open were analysed

Could the authors comment as to why patients were lost to follow up

How was the p-value for postoperative pain derived given that data was presented for 4 possible outcomes of no pain/mild/moderate/severe?

Figure 3 Could the axes be labelled? or any other information that would aid the interpretation of this figure

Line 193-197 It is apparent that the respiratory parameters have achieved statistical significance but it is less apparent that a respiratory rate difference of 1 or oxygen saturations of 97% vs 99% is of clinical significance

Discussion

Could the authors give more background as to any links between SIRS or IL-6 and clinical outcomes?

Could the authors comment as to why laparoscopic cholecystectomies were the target population of this study given that laparoscopic surgery intuitively seems to be less associated with SIRS

My understanding is that IL-6 and CRP are closely linked. Could the authors comment as to why the attenuation was seen in one but no the other.

Could the authors comment to their perceived limitations of their study

Line 225 - Could the authors clarify what is meant by 'soon'. This statement appears key to the whole study. Soon as in we are awaiting this data?

Line245-251 - This paragraph appears reductive of inflammation around surgery and might be reflective of the reference specifically looking at vagus nerve stimulation

Line 295 - 'quality' feels like a broader concept that just rate and saturation

Conclusions

Line 306 - 'better clinical outcomes' is a broad conclusion

Line 307 - 'earlier hospital discharge and decreased hospital costs' can this be concluded from this study

References

35 - Is this 1991 paper the best reference to be used?

Author Response

Author's Notes:

     We are glad to know that, based on the advice received, our manuscript entitled “Effect of Dexmedetomidine on Immunomodulation and Pain Control in Videolaparoscopic cholecystectomies: A Randomized, Two-Arm, Double-Blinded, Placebo-Controlled Trial" has merit for publication. We thank you for your time spent evaluating our manuscript and for the valuable input given to improve it.

       We send a updated version for your appreciation.

Round 2

Reviewer 2 Report

Thank you for the response to reviewers and changes to the manuscript. The amended manuscript has adequately addressed the comments previously provided. 

I would like to make the following minor comments to the revised version.

Line 49 - pneumoperitoneum arguably has more cardiorespiratory implications. Such as increased afterload, affects on venous return, vagal tone, diaphragmatic splinting and increased ventilatory pressures.

Line 55 - This reads as if bronchoconstriction is part of the physiology of tracheal intubation. I would argue that it is a separate pathology that can, but not necessarily occur with tracheal intubation.

Line135 - Could the authors add in the timing of the dexmedetomidine infusion? It would aid in the interpretation of the haemodynamic variables such as reduced HR and BP on induction. Is the bolus complete or started with time 0"?

Also, the previous version made reference to TCI modelling, was this not used? 

Was there a way in which the protocol ensured complete return of neuromuscular function? Was neuromuscular monitoring used or routine reversal given? This would have implications on the respiratory function reported.

Author Response

   We are glad to know that, based on the advice received, our manuscript entitled “Effect of Dexmedetomidine on Immunomodulation and Pain Control in Videolaparoscopic cholecystectomies: A Randomized, Two-Arm, Double-Blinded, Placebo-Controlled Trial" has merit for publication. We thank you for your time spent evaluating our manuscript and for the valuable input given to improve it.

     We send a updated version for your appreciation.
